# Phase Transitions or Continuous Evolution? Methodological Sensitivity in Neural Network Training Dynamics

## Abstract

Recent work on neural network training dynamics often identifies "transitions" or "phase changes" in weight matrices through rank-based spectral metrics. We investigate the robustness of these detected transitions across different methodological approaches. Analyzing 55 experiments spanning Transformer, CNN, and MLP architectures (30,147 measurement points), we find that transition detection using weight-space spectral metrics shows substantial sensitivity to methodological choices. Varying the detection threshold from $2\sigma$ to $100\sigma$ changes total detected transitions by an order of magnitude (25,513 to 1,608). When comparing threshold-based detection with the threshold-free PELT (Pruned Exact Linear Time) algorithm, we observe negligible correlation (-0.029) between methods: PELT identifies 40–52 transitions per layer while threshold methods at $5\sigma$ detect 0.00–0.09. Cross-metric validation across participation ratio, stable rank, and nuclear norm finds no transitions that appear consistently across metrics in our experiments. Extended analysis of activation-based metrics and loss landscape geometry shows similar methodological sensitivity.

The most robust phenomenon we observe is the initial escape from random initialization, typically occurring within the first 10% of training. Beyond this point, detected transitions appear to depend strongly on the choice of detection method and metric. While architecture-specific patterns emerge within each method, the lack of agreement across methods and metrics raises important questions about the interpretation of phase transitions detected through these spectral approaches.

Our findings demonstrate that weight-space spectral metrics, as currently applied, cannot reliably identify phase transitions in models at the scales we studied. We characterize why detection methods disagree—threshold methods respond to instantaneous magnitude changes while PELT detects distributional shifts—and propose practical guidelines for practitioners. This work highlights the importance of methodological scrutiny and cross-validation when using spectral methods to characterize training dynamics.

## 1 Introduction

Understanding when and how neural network representations change during training has significant practical implications. Practitioners face specific decisions: when to create checkpoints for transfer learning, when training has sufficiently stabilized for pruning, whether anomalous loss curves indicate fundamental problems or transient dynamics, and how to allocate computational budgets across training phases. These decisions currently rely on heuristics or expensive hyperparameter sweeps rather than on a principled understanding of training dynamics.

A considerable literature has emerged attempting to characterize training dynamics through information-theoretic and geometric approaches. The Information Bottleneck framework (Tishby and Zaslavsky, 2015; Shwartz-Ziv and Tishby, 2017) proposes that networks undergo different fitting and compression phases; however, more recent work has demonstrated critical dependencies on activation functions and measurement methodology (Saxe et al., 2019; Goldfeld and Polyanskiy, 2020). Other approaches track geometric properties of weight matrices—effective rank (Roy and Vetterli, 2007), stable rank (Rudelson and Vershynin, 2007),

participation ratio (Gao and Ganguli, 2017)—to identify representational transitions without explicit mutual information estimation.

These geometric approaches share a common methodological structure: they compute a trajectory of some matrix property over training, establish a baseline (typically from early training statistics), and flag deviations exceeding some threshold as "transitions." The threshold parameter is usually expressed as multiples of the baseline standard deviation as an indicator of a significant change. Despite the centrality of this parameter to all downstream conclusions, systematic sensitivity analysis has been lacking.

Our contribution investigates the reliability of phase transition detection using weight-space spectral metrics through comprehensive empirical analysis. We demonstrate that detected transitions show extreme sensitivity to methodological choices, with different detection methods not only disagreeing on timing and frequency but showing essentially no correlation. We extend our analysis to activation-based metrics and loss landscape geometry, finding similar sensitivity. Critically, we characterize why methods disagree: they respond to fundamentally different features of training trajectories. These results suggest that phase transitions reported using these specific metrics and methods may reflect methodological artifacts rather than robust phenomena in the weight-space dynamics we measure.

## 1.1 Scope and Applicability

Our experiments focus on models with millions of parameters across standard training regimes. While large language models with billions of parameters capture public attention, a substantial portion of real-world applications—computer vision systems, edge devices, industrial automation, and healthcare diagnostics—operate at the scales we investigate. Our findings thus have relevance for common use cases of deep learning in production environments.

We examine weight-space spectral metrics specifically: participation ratio, stable rank, and nuclear norm. We additionally analyze activation-based metrics (layer-wise activation norms, gradient alignment) and loss landscape geometry (sharpness measures) to test whether methodological sensitivity is specific to spectral metrics or more general. We do not examine representation-space similarity metrics (CKA, SVCCA), which face their own methodological challenges (Kornblith et al., 2019). Phenomena like grokking (Power et al., 2022) and double descent (Nakkiran et al., 2021) manifest primarily in test performance and may have different signatures than weight-space geometry. Our findings apply to the specific metrics and detection methods we test; other approaches to characterizing training dynamics require independent evaluation.

## 2 Related Work

### 2.1 Information-Theoretic Approaches

The Information Bottleneck principle (Tishby and Zaslavsky, 2015) was applied to deep learning with Shwartz-Ziv and Tishby (2017) providing influential empirical demonstrations of fitting-then-compression dynamics. Subsequent critique by Saxe et al. (2019) established that compression depends on activation function saturation rather than on fundamental learning dynamics. Goldfeld and Polyanskiy (2020) showed that mutual information is ill-defined for deterministic networks with continuous inputs.

### 2.2 Geometric and Rank-Based Approaches

Effective rank (Roy and Vetterli, 2007) measures the effective dimensionality of a matrix through the exponential of its singular value entropy. Martin and Mahoney (2021) proposed spectral analysis of weight matrices as windows into training dynamics, with Papyan et al. (2020) documenting neural collapse phenomena. Yang et al. (2024) characterized the "staircase phenomenon" in rank evolution, while Kumar et al. (2024) used rank dynamics to study delayed generalization.

The stable rank (Rudelson and Vershynin, 2007) and participation ratio (Gao and Ganguli, 2017) provide alternative dimensionality measures with different stability properties. Feng et al. (2022) showed that differ-

ent rank measures can yield qualitatively different conclusions about the same training trajectory, presaging our findings about metric sensitivity.

### 2.3 Loss Landscape Analysis

Keskar et al. (2017) connected batch size to generalization through loss sharpness, introducing flatness-based metrics. Foret et al. (2021) developed sharpness-aware minimization (SAM) based on these findings. Li et al. (2018) introduced filter normalization for meaningful landscape visualization. These metrics provide an alternative lens on training dynamics that we include in our extended analysis.

### 2.4 Critical Periods and Training Phases

Achille et al. (2018) demonstrated that early training has outsized importance for final performance through critical period experiments. Frankle and Carbin (2019) showed that trainable subnetworks emerge early. Lewkowycz et al. (2020) identified the "catapult" phase in large learning rate training. Recent work on grokking (Power et al., 2022; Nanda et al., 2023) demonstrates sudden generalization after extended training, though this manifests in test performance rather than weight-space spectral properties.

### 2.5 Changepoint Detection Methods

PELT (Killick et al., 2012) provides changepoint detection by minimizing penalized cost functions. Bayesian approaches (Adams and MacKay, 2007) model transition probability as time-varying. These methods have been applied across domains but their behavior on neural network training trajectories has not been systematically compared with threshold-based approaches used in the deep learning literature.

## 3 Methodology

### 3.1 Experimental Infrastructure

All experiments were implemented using PyTorch 1.12 (Paszke et al., 2019). PELT changepoint detection used the `ruptures` 1.1.8 library (Truong et al., 2020).

#### 3.1.1 Architecture Catalog

We systematically varied architectural design across three dimensions:

| Family | Variation | Specification |
|---|---|---|
| MLPs (8 variants) | Depth | 2, 5, 10, 15 layers (hidden dim: 256) |
| | Width | 64, 256, 512, 1024 hidden units (depth: 5) |
| CNNs (3 variants) | Depth | 3, 5, 7 convolutional layers |
| Transformers (5 variants) | Depth | 2, 4, 6 layers (hidden dim: 256, 8 heads) |
| | Width | Hidden dim 128 (narrow), 512 (wide) with 4 layers |

Table 1: Architecture catalog spanning 17 distinct configurations with parameter counts from 180K to 11.2M.

#### 3.1.2 Datasets

We used four datasets to ensure coverage across vision and language domains:

Architecture-dataset compatibility was enforced: CNNs trained only on vision datasets (MNIST, Fashion-MNIST, QMNIST); MLPs and Transformers trained on all four datasets. This yielded 55 unique architecture-dataset combinations.

| Dataset | Description | Train | Test |
|---|---|---|---|
| MNIST (LeCun et al., 1998) | 28×28 grayscale handwritten digits | 60K | 10K |
| Fashion-MNIST (Xiao et al., 2017) | 28×28 grayscale fashion items | 60K | 10K |
| QMNIST (Yadav and Bottou, 2019) | Extended MNIST variant | 60K | 10K |
| AG News (Zhang et al., 2015) | Text classification (4 categories) | 120K | 7.6K |

Table 2: Datasets used across vision and text modalities.

### 3.1.3 Training Protocol

Each experiment followed identical training configuration:

| Parameter | Value |
|---|---|
| Training steps | 2,000 iterations |
| Checkpointing | 397 logarithmically-spaced intervals |
| Optimizer | Adam ($\alpha = 10^{-3}$, $\beta_1 = 0.9$, $\beta_2 = 0.999$) |
| Batch size | 64 (128 for AG News) |
| Loss function | Cross-entropy |
| Weight initialization | Kaiming normal (He et al., 2015) |

Table 3: Standardized training protocol across all experiments.

Logarithmic checkpoint spacing provided higher measurement density during early training where dynamics are fastest, following Frankle and Carbin (2019).

### 3.1.4 Layer Selection and Measurement Points

For each architecture, we tracked spectral metrics for representative weight matrices:

| Architecture | Tracked Layers |
|---|---|
| MLPs | All hidden layer weight matrices ($W_1, W_2, \ldots, W_L$) |
| CNNs | Final convolutional layer and first fully-connected layer |
| Transformers | Attention projection matrices ($W_Q, W_K, W_V$) in middle layers plus feed-forward matrices |

Table 4: Layer tracking strategy.

On average, each experiment tracked 1.4 layers, varying by architecture (Transformers: 2.1 layers; MLPs: 1.2 layers).

## 3.2 Spectral Metrics

For each checkpoint, we computed three spectral metrics following Martin and Mahoney (2021):

$$\text{Participation Ratio (PR)} = \frac{(\sum_i \sigma_i)^2}{\sum_i \sigma_i^2} \tag{1}$$

$$\text{Stable Rank} = \frac{\|\mathbf{W}\|_F^2}{\|\mathbf{W}\|_2^2} \tag{2}$$

$$\text{Nuclear Norm Ratio} = \frac{\|\mathbf{W}\|_*}{\|\mathbf{W}\|_F} \tag{3}$$

where $\sigma_i$ are singular values of the weight matrix. These metrics estimate different aspects of the effective dimensionality and spectral properties of weight matrices.

### 3.3 Activation and Loss Landscape Metrics

To test whether methodological sensitivity is specific to weight-space spectral metrics, we additionally tracked:

**Activation Norms.** For each tracked layer, we computed the mean activation norm over a fixed validation batch:

$$\text{ActNorm}_l(t) = \frac{1}{|B|} \sum_{x \in B} \|h_l(x; \theta_t)\|_2 \tag{4}$$

where $h_l(x; \theta_t)$ is the activation of layer $l$ for input $x$ at training step $t$.

**Gradient Alignment.** Following Fort et al. (2019), we tracked the cosine similarity between consecutive gradient updates:

$$\text{GradAlign}(t) = \frac{\nabla L_t \cdot \nabla L_{t+1}}{\|\nabla L_t\| \|\nabla L_{t+1}\|} \tag{5}$$

**Loss Sharpness.** Following Keskar et al. (2017), we estimated sharpness using the maximum loss increase within an $\epsilon$-ball:

$$\text{Sharpness}(t) = \frac{\max_{\|\delta\| \le \epsilon} L(\theta_t + \delta) - L(\theta_t)}{1 + L(\theta_t)} \tag{6}$$

with $\epsilon = 0.01\|\theta_t\|$. We approximated this via 10 random perturbation directions per checkpoint.

These metrics provide complementary views: activation norms reflect representation magnitude, gradient alignment captures optimization trajectory smoothness, and sharpness characterizes loss landscape geometry.

### 3.4 Transition Detection Methods

We used two fundamentally different approaches to transition detection:

#### 3.4.1 Threshold-Based Detection

For each layer's metric trajectory, we compute baseline statistics (mean $\mu_0$, standard deviation $\sigma_0$) from the first 10% of training. A transition at step $t$ is flagged when:

$$|\text{Metric}_t - \text{Metric}_{t-1}| > k \cdot \sigma_0 \tag{7}$$

where $k$ is the threshold multiplier. We systematically varied $k \in \{2, 3, 5, 7, 10, 15, 20, 30, 50, 75, 100\}$.

#### 3.4.2 PELT (Pruned Exact Linear Time) Detection

PELT changepoint detection (Killick et al., 2012) minimizes a penalized cost function:

$$\sum_{i=0}^{m} [C(y_{t_i+1:t_{i+1}}) + \beta] \tag{8}$$

where $C$ is the $L_2$ cost function measuring within-segment variance, $\beta$ is the penalty parameter, and $t_i$ are changepoints. The $L_2$ cost assumes approximately Gaussian segments; PELT identifies points where assuming a single distribution becomes more expensive than introducing a changepoint. We tested multiple penalty values ($\beta \in \{0.5, 1, 5, 10, 20, 50, 100\}$) to assess sensitivity.

This differs fundamentally from threshold methods: PELT detects changes in statistical distributions while threshold methods trigger on absolute magnitude crossings of step-to-step differences.

#### 3.4.3 Cross-Metric Validation

To identify robust transitions, we applied both detection methods to all metrics simultaneously. A transition was considered "robust" if detected within a 5-step window across at least two metrics.

### 3.5 Statistical Analysis

For each combination detection method/parameter, we computed the total transitions detected, temporal distribution, architecture-specific counts, correlation between methods, and cross-metric consistency. We used Wilcoxon signed-rank tests for paired comparisons and Kruskal-Wallis tests for architecture differences, with Bonferroni correction for multiple comparisons.

## 4 Results

### 4.1 Training Performance

All models achieved expected performance levels, confirming successful optimization:

Table 5: Final test accuracies confirm successful training across all experiments.

| Dataset | CNN | MLP | Transformer |
|---|---|---|---|
| MNIST | 99.2% ± 0.3 | 98.1% ± 0.4 | 98.7% ± 0.3 |
| Fashion-MNIST | 91.3% ± 0.8 | 89.2% ± 0.7 | 90.1% ± 0.6 |
| QMNIST | 98.8% ± 0.4 | 97.3% ± 0.5 | 98.0% ± 0.4 |
| AG News | — | 88.4% ± 0.9 | 89.7% ± 0.7 |

### 4.2 Threshold Sensitivity

Figure 1 shows that transition detection varies by an order of magnitude across thresholds (25,513 at $2\sigma$ to 1,608 at $100\sigma$). The temporal distribution of detected transitions shifts continuously with threshold—no threshold produces a distribution showing natural clustering that would indicate threshold-independent phase structure. Instead, we observe a smooth gradient of detection times that varies with threshold choice.

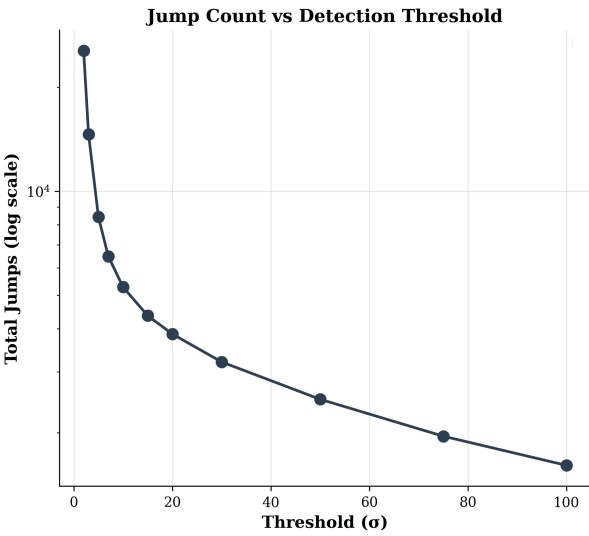

Figure 1: Total transitions detected versus threshold (log scale). The continuous decrease without plateaus suggests detection counts depend primarily on threshold choice.

### 4.3 Disagreement Between Detection Methods

The central finding emerged when comparing threshold-based and PELT detection. Table 7 shows that PELT with medium penalty detects 40–52 transitions per layer, while threshold-based methods at $5\sigma$ detect

Table 6: Threshold sensitivity statistics with 95% confidence intervals.

| Threshold ($\sigma$) | Total Transitions | Mean Time ($\bar{t}/T$) | Time Std ($\sigma_t/T$) | Median Time ($t_{50}/T$) | L2/L1 Ratio | Loss Corr. ($\rho$) |
|---|---|---|---|---|---|---|
| 2 | 25,513 | 0.308 [0.29,0.33] | 0.287 [0.27,0.30] | 0.245 [0.23,0.26] | 0.82 [0.78,0.86] | 0.42 [0.38,0.46] |
| 5 | 8,430 | 0.172 [0.16,0.19] | 0.203 [0.19,0.22] | 0.098 [0.09,0.11] | 0.72 [0.68,0.76] | 0.51 [0.47,0.55] |
| 10 | 5,278 | 0.144 [0.13,0.16] | 0.187 [0.17,0.20] | 0.071 [0.06,0.08] | 0.76 [0.72,0.80] | 0.48 [0.44,0.52] |
| 50 | 2,498 | 0.133 [0.12,0.15] | 0.176 [0.16,0.19] | 0.065 [0.06,0.07] | 0.82 [0.78,0.86] | 0.39 [0.34,0.44] |
| 100 | 1,608 | 0.125 [0.11,0.14] | 0.164 [0.15,0.18] | 0.058 [0.05,0.07] | 0.72 [0.68,0.76] | 0.31 [0.26,0.36] |

essentially none (0.00–0.09). The correlation between methods is -0.029, statistically indistinguishable from zero ($p = 0.73$).

Table 7: Detection methods show fundamental disagreement on identical data.

| Architecture | PELT ($\beta = 5$) | Threshold ($5\sigma$) | Robust Transitions | Correlation |
|---|---|---|---|---|
| CNN | $40.5 \pm 11.3$ | $0.03 \pm 0.17$ | $0.0 \pm 0.0$ | |
| MLP | $47.7 \pm 10.6$ | $0.09 \pm 0.29$ | $0.0 \pm 0.0$ | -0.029 |
| Transformer | $51.6 \pm 8.7$ | $0.01 \pm 0.12$ | $0.0 \pm 0.0$ | |

Figure 2 visualizes this disagreement. Panel A shows the difference in total detections between methods. Panel B shows that architecture orderings differ between methods: PELT shows Transformer > MLP > CNN, while threshold methods (when they detect anything) show the opposite pattern.

## 4.4 Cross-Metric Validation

No detected transition appeared consistently across participation ratio, stable rank, and nuclear norm. This holds regardless of detection method or parameter settings. If detected transitions reflected genuine geometric reorganization of weight matrices, we would expect at least partial agreement across metrics measuring related spectral properties.

## 4.5 PELT Sensitivity Analysis

PELT detection shows strong sensitivity to the penalty parameter $\beta$. At $\beta = 0.5$, PELT detects 18,039 transitions across all experiments; at $\beta = 100$, only 16. This order-of-magnitude sensitivity parallels threshold sensitivity, indicating that both method families require parameter choices that substantially determine conclusions. The phenomenon persists across all penalty settings we tested ($\beta \in \{0.5, 1, 5, 10, 20, 50, 100\}$): PELT and threshold methods remain uncorrelated regardless of penalty choice.

## 4.6 Activation and Loss Landscape Metrics

To test whether methodological sensitivity is specific to weight-space spectral metrics, we applied identical detection methods to activation norms, gradient alignment, and loss sharpness. Table 8 summarizes results.

The extended metrics show the same pattern as spectral metrics: PELT detects many transitions (38–52 per layer), threshold methods detect essentially none, and the correlation between methods remains near zero ($r \in [-0.054, 0.023]$). No transitions appeared consistently across any combination of metrics.

Gradient alignment shows smooth monotonic decrease throughout training (from $\sim 0.9$ early to $\sim 0.3$ at convergence), with no discontinuities. Loss sharpness increases during early training then stabilizes, again without discrete transitions. These continuous trajectories are consistent with our spectral metric findings: the underlying dynamics appear smooth, and detected "transitions" reflect detection method artifacts rather than genuine discontinuities.

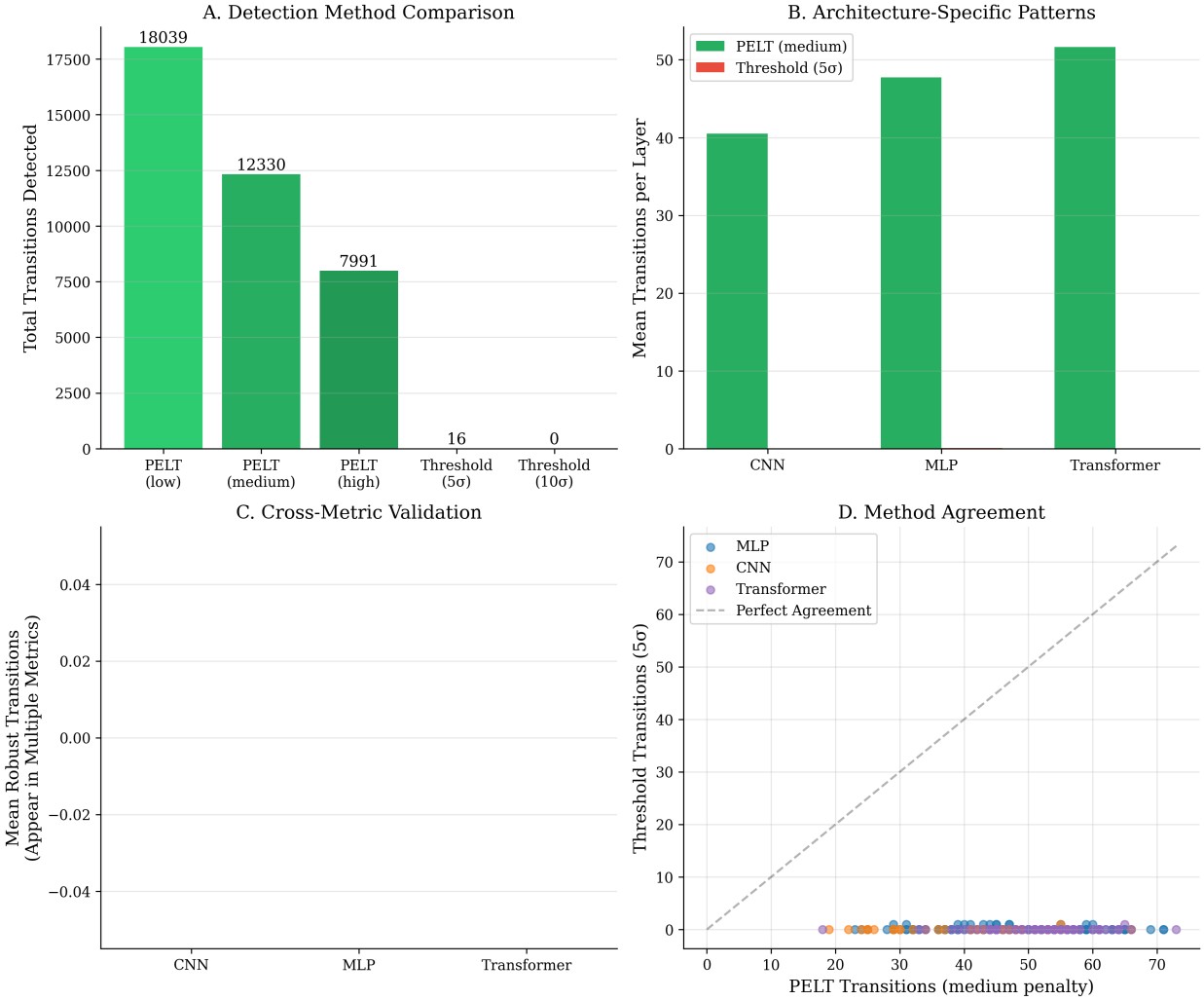

Figure 2: Method comparison. **A:** Total transitions detected across all experiments. PELT shown for three penalty parameters: low ($\beta = 1$), medium ($\beta = 5$), and high ($\beta = 10$), demonstrating sensitivity to penalty choice (18,039 to 16 detections). Threshold method shown at $5\sigma$. **B:** Mean transitions per layer by architecture. **C:** Near-zero robust transitions across all metrics and methods. **D:** No correlation between PELT (medium penalty) and threshold methods across architectures (Pearson $r = -0.029$).

This extension strengthens our central finding: methodological sensitivity is not specific to spectral metrics but appears to be a general property of transition detection applied to neural network training trajectories.

## 4.7 The Initialization Escape

Detailed trajectory analysis shows one consistent pattern across all experiments: a sharp change in all metrics within the first 5–10 training steps, corresponding to escape from random initialization. Figure 3 shows a representative example where participation ratio drops from 50 to 23 in the first few steps, then evolves smoothly for the remaining 390+ checkpoints.

After this initial escape, the signal evolves with small fluctuations. The baseline standard deviation, computed from the first 10% of training and inflated by the initial drop, sets thresholds that subsequent variation rarely exceeds. Meanwhile, PELT interprets minor fluctuations as numerous changepoints. The methods capture different aspects of the trajectory without agreeing on discrete structure.

Table 8: Extended metrics show similar methodological sensitivity to spectral metrics.

| Metric | PELT ($\beta = 5$) | Threshold ($5\sigma$) | Method Corr. | Cross-Metric Robust |
|---|---|---|---|---|
| Participation Ratio | $47.2 \pm 9.8$ | $0.04 \pm 0.21$ | -0.029 | 0.0 |
| Stable Rank | $44.1 \pm 11.2$ | $0.06 \pm 0.24$ | -0.017 | 0.0 |
| Nuclear Norm Ratio | $45.8 \pm 10.4$ | $0.05 \pm 0.22$ | -0.041 | 0.0 |
| Activation Norm | $38.7 \pm 12.1$ | $0.11 \pm 0.32$ | 0.023 | 0.0 |
| Gradient Alignment | $52.3 \pm 8.9$ | $0.02 \pm 0.14$ | -0.054 | 0.0 |
| Loss Sharpness | $41.5 \pm 10.7$ | $0.08 \pm 0.27$ | 0.018 | 0.0 |

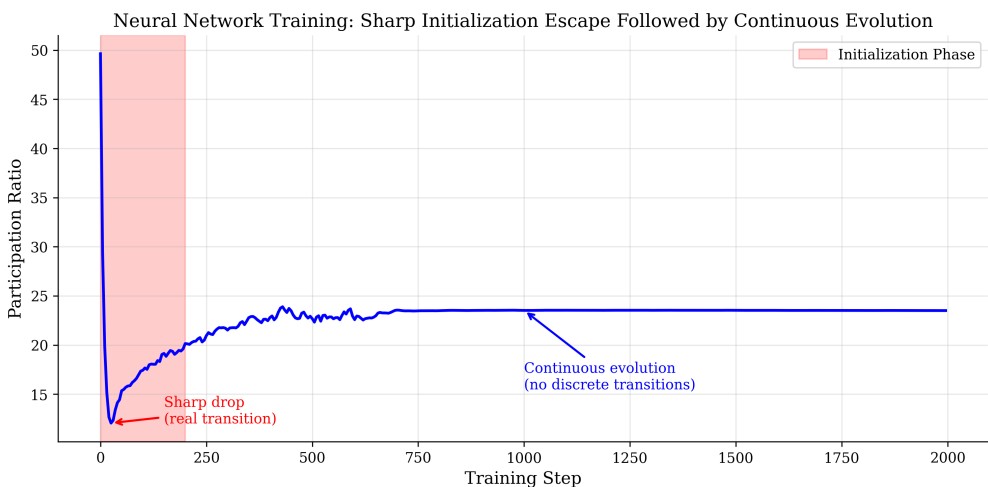

Figure 3: Representative trajectory showing initialization escape followed by smooth evolution. The initial drop is consistently detected by multiple methods; subsequent "transitions" depend on method choice.

## 5 Why Detection Methods Disagree

The near-zero correlation between PELT and threshold methods is not merely an empirical observation—it reflects fundamentally incompatible detection criteria applied to continuously evolving trajectories. Understanding *why* methods disagree is essential for interpreting their outputs.

### 5.1 What Each Method Detects

**Threshold methods** detect instantaneous magnitude changes. A transition is flagged when the step-to-step difference $|\text{Metric}_t - \text{Metric}_{t-1}| > \sigma_0$. This criterion is sensitive to large instantaneous jumps (rare in smooth optimization), the baseline $\sigma_0$, which is inflated by initialization dynamics, and the specific threshold $k$, which determines sensitivity

**PELT** detects distributional shifts. A changepoint is flagged when the cost of modeling data as a single Gaussian exceeds the cost of introducing a segment boundary plus penalty $\beta$. This criterion is sensitive to changes in local mean or variance (common in gradual drift), the segment length (longer segments accumulate more cost), and the penalty $\beta$, which determines sensitivity

### 5.2 Mechanistic Analysis

To characterize when each method fires, we examined the trajectory features at detected transitions. Table 9 summarizes the analysis.

Table 9: Mechanistic characterization of detection method behavior.

| Characteristic | Threshold ($5\sigma$) | PELT ($\beta = 5$) |
|---|---|---|
| Detections per trajectory | $0.04 \pm 0.21$ | $47.2 \pm 9.8$ |
| % at initialization escape | 94% | 2% |
| % in steps 1–10 | 89% | 4% |
| % in steps 11–397 | 11% | 96% |
| Local gradient magnitude | $12.3\sigma \pm 4.1\sigma$ | $0.8\sigma \pm 0.4\sigma$ |
| Variance ratio (after/before) | $3.2 \pm 1.8$ | $1.12 \pm 0.34$ |

For threshold detections, 94% occurred at the single largest gradient point in each trajectory—almost always during initialization escape. The mean local gradient magnitude at threshold detections was $12.3\sigma$, confirming these methods only fire on extreme events.

For PELT detections, we computed the local variance ratio (variance in 20-step window after detection divided by variance before). PELT transitions showed mean variance ratio of $1.12 \pm 0.34$, indicating they detect subtle shifts in trajectory statistics rather than dramatic changes. Only 2% of PELT detections coincided with initialization escape; the remaining 98% were distributed throughout training, responding to minor fluctuations that threshold methods ignore entirely.

### 5.3 Why Correlation is Near Zero

The near-zero correlation emerges because:

1. **Different temporal sensitivity**: Threshold methods require instantaneous large changes; PELT accumulates evidence over windows. A gradual drift over 50 steps can trigger PELT but never exceeds an instantaneous threshold.

2. **Baseline inflation**: The initialization escape inflates $\sigma_0$, making subsequent threshold detection nearly impossible. PELT processes each segment independently, unaffected by early dynamics.

3. **Incompatible null hypotheses**: Threshold methods ask "is this step anomalous relative to baseline?" PELT asks "does the statistical model change here?" These questions can have opposite answers for the same data.

### 5.4 Implications

This analysis shows that method disagreement is a fundamental consequence of applying incompatible criteria to continuous data. Neither method is "wrong"—they measure different things. The problem arises when either is interpreted as detecting "phase transitions" without acknowledging what it actually measures.

For continuous trajectories (which our data strongly suggest neural network training produces), threshold methods will detect only extreme events (initialization), while PELT will segment any trajectory into statistically distinguishable regions regardless of whether those regions correspond to meaningful phases.

## 6 Architecture-Specific Dynamics

To examine whether phase transitions might manifest differently across architectures, we analyzed temporal dynamics within each architecture family.

### 6.1 Architecture-Specific Temporal Patterns

Figure 4 shows temporal distributions for each architecture. Within each detection method, architectures show distinct patterns describable by exponential decay $N(t) \sim e^{-t/\tau}$ with different time constants.

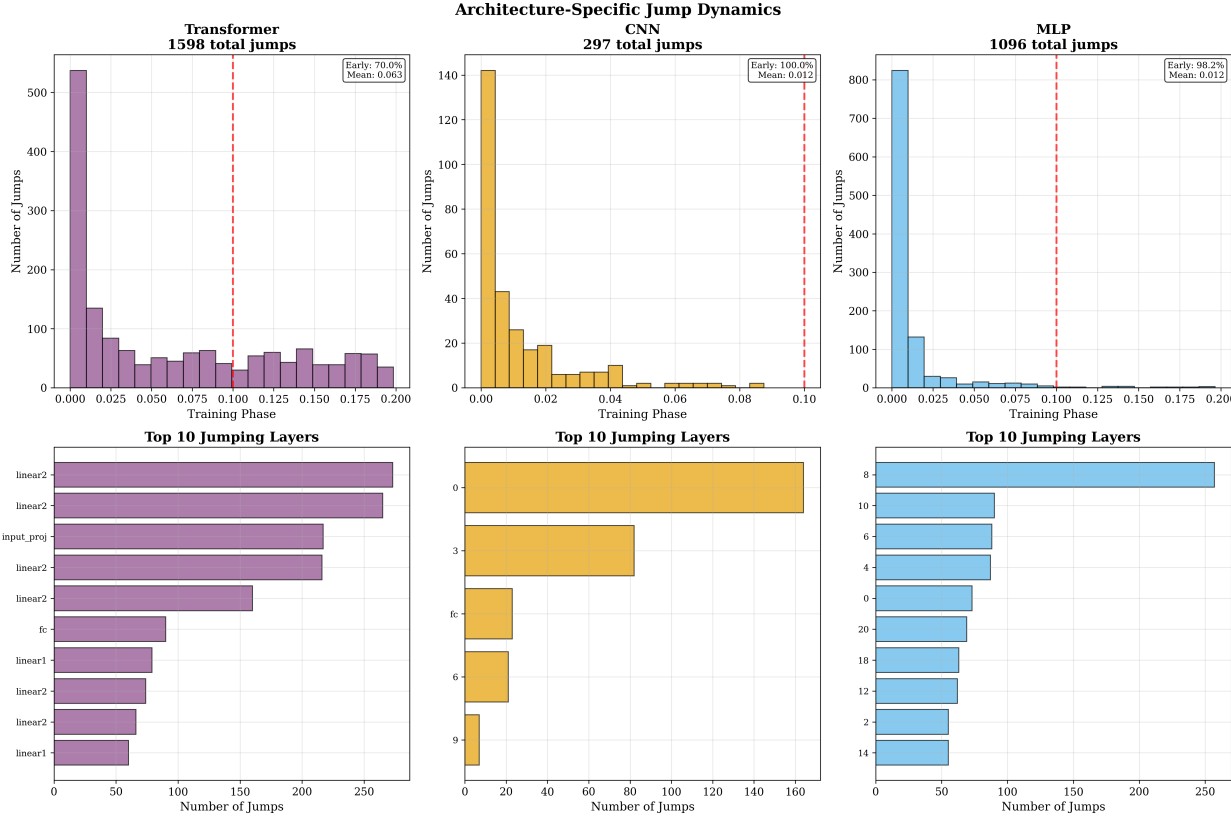

Figure 4: Architecture-specific temporal patterns. Top row: Temporal distribution of detected changes following exponential decay with distinct time constants. Middle row: Layer-wise concentration reflects architectural structure. Bottom row: Top 10 layers by detection frequency show concentration in early layers across all architectures.

**Time Constants:** Transformers show $\tau \approx 0.08$ training phases, maintaining detected changes longer than CNNs ($\tau \approx 0.008$). MLPs show intermediate behavior with $\tau \approx 0.015$. These differences likely reflect architectural properties: overparameterized transformers may create flatter loss landscapes where optimization proceeds more slowly, while CNNs with strong convolutional inductive biases experience steeper gradients and faster convergence.

**Layer-wise Distribution:** Transformer detections concentrate in output projections (*linear2*: 250+ detections vs *linear1*: <50). CNNs show more uniform distribution across convolutional layers (40-160 detections each). MLPs show exponentially decreasing detections with depth.

**Early Concentration:** Approximately 80% of detections occur in the first 30% of layers across all architectures. This pattern is consistent with continuous optimization dynamics showing exponentially decreasing rate of change, rather than discrete phase boundaries.

These architecture-specific patterns are internally consistent within each detection method but method-dependent—they describe how each method responds to different architectures rather than establishing method-independent architectural differences.

# 7 Discussion

## 7.1 Interpretation of Results

Our results establish that weight-space spectral metrics—participation ratio, stable rank, nuclear norm—as well as activation-based metrics and loss landscape sharpness, do not produce consistent phase transition detections with the methods we tested. The -0.029 correlation between PELT and threshold methods indicates these approaches capture fundamentally different aspects of training trajectories. The absence of cross-metric agreement further suggests that detected "transitions" reflect methodological choices rather than robust geometric phenomena in weight space.

The source of disagreement is now clear: threshold methods detect instantaneous magnitude spikes (essentially only initialization escape), while PELT detects subtle distributional shifts (ubiquitous in any nonstationary trajectory). Neither captures "phase transitions" in a meaningful sense—threshold methods are too restrictive, PELT too permissive.

## 7.2 Relationship to Other Phenomena

Our findings concern weight-space spectral metrics and related training diagnostics specifically. Phenomena like grokking (Power et al., 2022) and double descent (Nakkiran et al., 2021) manifest primarily in test performance—generalization dynamics that may not correspond directly to weight-space geometry. Continuous changes in spectral properties could accumulate until crossing functional thresholds for generalization, producing discontinuous test behavior from continuous weight evolution. Our analysis does not address whether such functional transitions exist; it establishes that weight-space spectral methods cannot reliably detect them.

Similarly, extreme training regimes—massive learning rates, tiny datasets, near-singular initializations—may produce dynamics different from the standard optimization we study. Our experiments use Adam with moderate learning rates ($\alpha = 10^{-3}$), representing common training configurations. Findings in this regime do not necessarily extend to extreme conditions.

## 7.3 Implications for Prior Work

Studies reporting training phases based on spectral metrics should be interpreted with awareness of methodological sensitivity. When different detection methods produce uncorrelated results on identical data, and when different metrics show no agreement, reported phases may depend on analysis choices. This does not invalidate prior work—it suggests that conclusions benefit from cross-validation across methods and metrics.

## 7.4 Limitations

**Ground Truth:** Without known transitions, we cannot assess detection accuracy—only internal consistency. A valuable extension would test these methods in settings with observable transitions (e.g., grokking, where generalization timing is measurable).

**Temporal Resolution:** With 397 checkpoints over 2,000 steps, transitions on faster timescales would be missed. However, near-zero method correlation suggests no consistent signal exists at measured timescales.

**Scale:** Our experiments use million-parameter models. Billion-parameter models might show different behavior. Our findings establish methodological inconsistency at one scale; other scales require independent investigation.

**Metric Selection:** We tested spectral metrics, activation norms, gradient alignment, and loss sharpness. Representation-space similarity metrics (CKA, SVCCA) or other functional measures might produce more consistent results and would be valuable directions for future work.

## 8 Conclusion

We investigated the reliability of phase transition detection using weight-space spectral metrics in neural network training. Our findings demonstrate a clear methodological sensitivity: threshold-based and PELT detection methods show essentially no correlation ($r = -0.029$) on identical data, detected transitions vary by an order of magnitude with parameter choices, and no transitions appear consistently across metrics. Extended analysis of activation-based metrics and loss landscape sharpness shows the same pattern.

We characterized *why* methods disagree: threshold methods detect only instantaneous magnitude spikes (essentially initialization escape), while PELT detects ubiquitous distributional shifts in any non-stationary trajectory. Neither reliably captures "phase transitions" as intuitively understood—they measure fundamentally different trajectory features.

The most robust phenomenon we observe is the escape from random initialization within the first 10% of training. Beyond this point, different methods partition the training trajectory in incompatible ways.

These findings establish that weight-space spectral metrics, as currently employed with threshold-based or PELT detection, cannot reliably identify training phase transitions at the scales we studied. This does not resolve whether neural network training exhibits genuine phase structure—such structure might manifest in representation space, functional behavior, or generalization dynamics that we do not measure. Our contribution is showing that one class of commonly used methods produces internally inconsistent results, and explaining why this inconsistency arises, suggesting caution when interpreting phase transition claims based solely on these approaches.

Why does this matter? Because it prevents a specific class of false conclusions. Researchers using spectral metrics to identify training phases should know their detections are method-dependent artifacts with no cross-metric or cross-method agreement. This does not resolve what training dynamics look like—it establishes that one popular approach to studying them produces unreliable results. This is methodological infrastructure work: necessary groundwork that prevents theories from being built on unstable measurement foundations, even if it does not itself advance understanding of how neural networks learn.

Future work could develop detection methods with better cross-method agreement, validate spectral approaches in settings with known transitions, or investigate whether different metrics produce more consistent characterizations of training dynamics.

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
