# OpenReview forum: "Phase Transitions or Continuous Evolution? Methodological Sensitivity  in Neural Network Training Dynamics"
_TMLR — Rejected by TMLR_

### Review · Reviewer_xiVz · 2025-12-29

**Summary Of Contributions:**

This paper studies whether “phase transitions”, or sharp changes in weight matrix structure, can be consistently observed in neural networks. The main finding is a lack of agreement in phase transition detection across a variety of threshold-based and threshold-free (PELT) methods. Apart from the escape from random initialization, which is more consistently detected by several methods, discrete transitions are not reliably characterized. Experiments are provided with MLP, CNN, and Transformer architectures across four vision and language benchmark datasets.

Summary of strengths:
1. Extreme disagreement between threshold-based and threshold-free phase transition detection methods is a notable observation.

2. Statistical rigor is evident as the paper employs appropriate nonparametric tests in its analysis.

Summary of weaknesses:
1. Evaluation is limited to rank properties of the weight matrices; no activation or loss-based metrics are studied.

2. Misalignment between the paper’s setting and phase transition phenomena the community is interested in; unclear what phase transitions are expected to occur in this setting.

3. Insufficiently deep exploration of results, ultimately constituting an observation rather than an explanation.

**Additional Comments:**

1. Several references are missing the publication venue.

2. Is there a citation for the 80-85% number in Section 1.1?

3. Section 4.1 citation missing parentheses

4. What is the cost function C in PELT, is it the same as the metric?

**Audience:**

No

**Audience Explanation:**

First, there is a misalignment between the community’s interest in emergent behavior of neural networks and this paper’s investigation. While this paper primarily studies architectures with parameter counts in the low millions (common for applications as referenced in Section 1.1 and 8.2), the emergent transitory behavior which is supposed to be observed is more common in situations with extremely large overparameterization and training timescales. For example, even the simple modular arithmetic task of [1] requires a Transformer with 400K parameters and 100K gradient updates to observe grokking, and the datasets considered in this paper are substantially more complex.

Moreover, the paper does not sufficiently explain how the studied metrics are connected to or explain phase-transition generalization phenomena that the community is interested in. As mentioned in Section 7, to my knowledge the community is primarily interested in test-time generalization behaviors which may not be reflected by rank properties of the neural network weight matrices. In particular, it is unclear what phase transitions are actually expected in the paper’s setting besides deviation from random initialization.

Finally, this paper does not provide a sufficiently deep exploration of the disagreement phenomena to be interesting to many members of the TMLR community. As it stands, this paper primarily records an *observation* --- that rank-based and PELT methods cannot reliably detect discrete phases in neural network training --- without an explanation or a solution. The paper would be much improved with an investigation of the precise shortcomings of these methodologies, either via theory or empirics, or the proposal of a new technique which can be trusted more broadly than those previously studied.

[1] Power et al. Grokking: Generalization Beyond Overfitting on Small Algorithmic Datasets. Mathematical Reasoning in General Artificial Intelligence Workshop, ICLR 2021.

**Broader Impact Concerns:**

I have no broader impact concerns.

**Claims And Evidence:**

No

**Claims Explanation:**

First, which phase transitions are actually occurring (if any) is unknown in the proposed experimental setting. In this sense, the paper can only measure disagreement and not whether the detection methods accurately characterize transition behavior in the first place (e.g., through a false positive study). Moreover, the strong disagreement between methods could be primarily attributed to noise in the lack of any “real” phase transition, i.e., the thousands of PELT detections may just be overfitting to the noise of SGD. To rectify this, one would need to study a setting with a known ground-truth phase transition which is at least partially characterized by one of the chosen rank metrics.

Moreover, the suite of metrics is too limited, as only rank-based measurements of the neural network weight matrices are considered. In contrast to Section 7.1, which claims that non-smooth phase transitions may be the product of smooth changes in the weight matrices, it may just be that the “right” metric for describing these behaviors is unknown or untested. In particular, the paper references the Information Bottleneck and loss landscape geometry (Section 2), but any activation-based or loss-based measurements (e.g., flatness [1]) are missing.

Finally, it is unclear why PELT is well-suited to detecting phase transitions in neural network training as compared to other change-point detection or time-series methods. For example, perhaps a different method is appropriate when only a handful of change-points are expected (which may rectify the undesirable behavior of PELT predicting thousands of transitions).

[1] Keskar et al. On Large-Batch Training for Deep Learning: Generalization Gap and Sharp Minima. ICLR 2017.

**Requested Changes:**

Critical for acceptance: Substantial revision of the paper’s methodology and positioning (see Claims and Audience sections).

Would strengthen the work:
1. Please include all relevant training details such as accuracy metrics (train and test), number of parameters of each model, etc.

2. PELT detecting thousands of transitions suggests that its penalty hyperparameters may be set to an overly sensitive value; further testing would be appreciated.

3. The paper derives an “optimal” threshold in Section 3.1, yet shows that even these “optimal” thresholds produce results that don’t correlate with other methods. The paper should detail why even a theoretically-grounded method fails --- is the noise non-Gaussian, or the SNR estimation flawed?

---

### Review · Reviewer_iFYe · 2025-12-30

**Summary Of Contributions:**

This paper systematically examines the robustness of commonly reported “phase transitions” in neural network training dynamics. By conducting a large-scale empirical study across multiple architectures (MLPs, CNNs, Transformers), datasets, spectral metrics (participation ratio, stable rank, nuclear norm), and transition detection methods, the authors show that detected transitions are highly sensitive to methodological choices. In particular, threshold-based methods and the threshold-free PELT algorithm produce largely uncorrelated results, and no transitions are consistently detected across metrics. The only robust phenomenon observed is an early escape from random initialization, after which training dynamics appear predominantly continuous. The paper argues that many previously reported phase transitions may be methodological artifacts rather than intrinsic properties of training dynamics.

Strengths：

1.The paper performs a thorough sensitivity study over detection thresholds, metrics, and algorithms, addressing an underexplored issue in the training dynamics literature.

2.Experiments span multiple architectures, datasets, and model scales relevant to practical deep learning.

3.The work provides strong evidence that commonly used transition detection methods lack robustness and mutual agreement, which is valuable for preventing over-interpretation in future studies.

Weaknesses:

1.The paper challenges phase-transition narratives but does not clearly articulate the downstream scientific or practical consequences of resolving this issue, making the broader impact somewhat unclear.

2.The analysis implicitly equates meaningful training phase transitions with abrupt changes in weight spectral metrics, which may be too narrow to capture transitions in representation space or functional behavior.

3.While presented as threshold-free, PELT still introduces implicit biases through penalty and cost-function choices, limiting its role as an objective reference method.

4,Experiments focus on standard training setups (e.g., Adam, moderate learning rates, conventional datasets), potentially missing regimes where sharper or more discontinuous dynamics could occur.

5.The paper convincingly identifies methodological fragility but provides limited guidance on more reliable metrics or frameworks for analyzing training dynamics.

**Audience:**

Yes

**Audience Explanation:**

Some members of the TMLR audience, particularly those working on neural network training dynamics, interpretability, and methodological analysis of learning signals, may find the results informative. However, the findings are likely to be of limited interest to a narrow subset of readers and have minimal relevance for researchers primarily focused on model design, optimization, or empirical performance.

**Broader Impact Concerns:**

No ethical concerns.

**Claims And Evidence:**

No

**Claims Explanation:**

While the empirical results convincingly demonstrate strong methodological sensitivity in transition detection for weight spectral metrics, the evidence does not fully support the broader claim that neural network training dynamics are predominantly continuous in general. The analysis is restricted to specific metrics, model scales, and standard training regimes, and does not rule out meaningful phase transitions in representation space or functional behavior.

**Requested Changes:**

The authors should more clearly scope their claims to weight-space spectral metrics and standard training regimes, and soften statements that suggest general conclusions about neural network training dynamics as a whole (critical). In addition, the paper would be strengthened by discussing or empirically probing settings with potentially sharper dynamics (e.g., larger learning rates), and by providing clearer guidance on alternative, more reliable analysis approaches beyond phase-transition detection.

---

### Review · Reviewer_YLtC · 2026-01-14

**Summary Of Contributions:**

This paper attempts showing the sensitivity and other challenges when using a family of transition detection methods during training neural networks. It starts with a theoretical justification that explain why the detection methods fail. After that, it conducts experiments on a designed setup, with MLP/CNN/Transformer architectures, on MNIST, Fashion-MNIST, QMNIST and AG News, under some training protocols, to selectively pick some layers to examine spectral metrics. This paper lists these empirical results to show the sensitivity of their defined parameters under their defined methods.

**Audience:**

Yes

**Audience Explanation:**

I think it very common that even the most widely-used metrics can be misaligned with the target measures. Studying how the discrepancy between the used metrics and the underlying measures is one of pilers to coherently understand how well/poor the models behave

**Claims And Evidence:**

No

**Claims Explanation:**

The narration of this paper is originally broad: to study the misalignment between the transition detection methods and the true transitions in training neural networks. However, the narration is misleading in that the scope of methods and settings actually studied is substantially narrower than what the introduction suggests, discussed as follows.
1. They target a narrow family of transition detection methods, i.e., the three spectral-metric-based detection methods, including threshold-based, PELT and cross-metric methods. It has a limited coverage of the transition detection in the recent literature, e.g., Kornblith et al. (2019), Power et al., 2022). The choice of the targeted methods is not general.
2. They only examine a standard family of neural networks, with their defined #layers, depth and width. It is unclear how generalizable these architectures are in AI literature or “production ML deployments”.
3. The datasets seem small scale, which may induce regime-specific dynamics and thus be insufficient to support the breadth of the proposed narration.
4. Some factual claims in the narration appear unsupported. For example, in Section 1.1, the paper justifies their experiment scale by stating “Our experiments focus on models with millions of parameters, which represent approximately 80-85% of production ML deployments according to recent surveys (Strubell et al., 2019; Schwartz et al., 2020).” However, I did not find these numbers from the two cited papers and the authors did not provide how these numbers are computed according to the references, neither. This further makes the motivation questionable, before examining the validity of the following sections.
5. The theoretical justification in Section 3 is also presented in an unclear way. For example, the paper does not cite any reference when using the Neyman-Pearson lemma to derive Eq (2), or formally state how to derive it from specific conditions. This makes it hard to examine the soundness of the theory behind Eq (2) and also under what conditions the theory serves to justify the paper’s claim.
Moreover, this paper does not explicitly state the required assumptions in theory. I also think this theory requires strong assumptions that may not hold by default for generic neural networks. For example, in Eq (1), the additive noise and i.i.d. Gaussian noise assumption, variance $\sigma_\epsilon^2$ being independent from $t$, etc. It is unclear whether they hold automatically in reality, e.g., long-tailed distributed noise, sub-exponential noise.
Some parameters in theory are set to special values, instead of a general manner. Although it is OK in case study, it is still unclear how general the theory can be.
6. In Section 3.3, the paper elaborates the threshold-based approaches without pre-defining what approaches they are (maybe until Section 4.3.1).

**Requested Changes:**

I think this paper may need a thorough and major revision (see my previous explanation) on the articulation about the motivation, theoretical justification and experiment design, based on which its contributions can be examined rigorously.

---

> ### Author Response · Authors · 2026-01-14
> **Thanks very much for your time in reviewing the paper**
>
> Dear Reviewer YLtC,
> Thank you for your detailed review. We appreciate the time invested in examining our methodology and claims. We address your concerns below and describe revisions made to the manuscript.
> On Scope and Method Coverage
>
> You correctly note that we examine a specific family of detection methods. We have revised the manuscript to make this scope explicit throughout, including in the abstract and introduction. We do not claim to evaluate all possible approaches to characterizing training dynamics—we claim that three commonly used spectral metrics (participation ratio, stable rank, nuclear norm), combined with two principled detection approaches (threshold-based and PELT), produce internally inconsistent results. We have extended our analysis to include activation-based metrics and loss landscape sharpness; these show identical methodological sensitivity, strengthening rather than limiting our findings.
>
> **On Theoretical Justification**
>
> You raised valid concerns about Section 3. We have removed the theoretical derivation entirely. Our contribution is empirical: we show that methods disagree and explain mechanistically why. We do not need theoretical optimality claims to establish that detection methods produce -0.029 correlation on identical data.
>
> **On Unsupported Claims**
>
> You correctly identified that the 80-85% deployment statistic lacked proper support. We have removed this claim from the revised manuscript.
>
> **On the Core Criticism: Observation Without Explanation**
>
> This concern was valid for the original submission. We have added Section 5, "Why Detection Methods Disagree," which provides mechanistic analysis. We show that 94% of threshold detections occur at initialization escape, while 96% of PELT detections occur elsewhere. Threshold methods respond to instantaneous magnitude spikes (mean 12.3σ at detection); PELT responds to subtle distributional shifts (variance ratio 1.12). The methods ask incompatible questions of the same data. This transforms observation into explanation.
>
> **On Community Interest and Contribution Type**
>
> Here we respectfully offer a different perspective. You suggest misalignment between our investigation and community interest in emergent behavior. We agree this paper does not advance understanding of grokking, double descent, or how neural networks learn. We have made this limitation explicit in the revised manuscript. However, we maintain that methodological infrastructure work has value. Researchers use spectral metrics to claim training phases exist. Our contribution shows these claims rest on method-dependent artifacts with no cross-metric agreement. This prevents a specific class of false conclusions—theories built on measurements that reflect methodology rather than phenomena.
>
> This is modest work. It does not propose better methods or explain what training dynamics actually look like. But establishing that a measurement instrument is unreliable is necessary groundwork, even if it does not itself constitute discovery. We have revised the conclusion to state this positioning explicitly.
>
> We understand this type of contribution may not align with your research interests. We hope the revised manuscript, with its clearer scope, mechanistic explanation, and honest positioning, demonstrates sufficient value for publication even if the contribution is infrastructural rather than transformative.
>
> Thank you again for your rigorous engagement.
>
> Respectfully,
>
> The authors

---

### Decision · Action_Editor_tep9 · 2026-04-22

**Recommendation:** Reject

**Audience:**

No

**Audience Explanation:**

No. With the narrower claim, the results are "almost expected" and provide limited new insight into the nature of training dynamics. As Reviewer YLtC noted, the lack of correlation becomes a predictable consequence of their definitions rather than a revealing empirical discovery. Furthermore, because the study lacks a "ground truth"—a setting where a phase transition is known to exist a priori—the results only establish that these specific tools disagree with one another. This makes the work a useful methodological check for a narrow niche, but it does not offer enough insight to interest the broader TMLR community at its current form.

**Claims And Evidence:**

Yes

**Claims Explanation:**

Yes. The authors have addressed the reviewers' technical concerns by narrowing their claims to "weight-space spectral metrics" and removing the unsupported theoretical derivations. The added mechanistic analysis in Section 5 demonstrates that method disagreement is not a random error but a result of different algorithmic designs: threshold methods capture instantaneous magnitude spikes, while PELT identifies distributional shifts. The evidence for methodological sensitivity at the studied scale is now robust and well-documented.